# What are the experiences and psychosocial needs of female survivors of domestic violence in Afghanistan? A qualitative interview study in three Afghan provinces

Anjuli Kaul ,[1] Lamba Saboor,[2] Ayesha Ahmad,[3] Jenevieve Mannell ,[2] Sharli Anne Paphitis,[1] Delan Devakumar[2]

¹Section of Women's Mental Health, King's College London, London, UK
²Institute for Global Health, University College London, London, UK
³St George's University of London, London, UK

**Correspondence to**
Anjuli Kaul;
anjuli.1.kaul@kcl.ac.uk

## ABSTRACT

**Objectives** This study aimed to qualitatively explore (1) the experiences of female survivors of domestic abuse and mental health problems in Afghanistan; (2) how female survivors of violence and abuse, male members of the community and service providers perceive and respond to mental health and domestic violence in Afghanistan and (3) the provision of mental health services for female survivors of violence and abuse in Afghanistan, including the barriers and challenges faced around accessing mental health services.

**Design** Qualitative interviews and framework thematic analysis.

**Setting** Kabul, Bamyan and Nangarhar in Afghanistan.

**Participants** 60 female survivors of domestic abuse, 60 male community members and 30 service providers who work with female survivors of domestic abuse.

**Results** Experiences of multiple and compounding traumatic experiences of violence, armed conflict, and complex and competing psychosocial concerns were common among the female survivor participants. All female survivor participants reported experiencing negative mental health outcomes in relation to their experiences of violence and abuse, which were further precipitated by widespread social stigma and gender norms. Support and service provision for female survivors was deemed by participants to be insufficient in comparison to the amount of people who need to access them.

**Conclusions** There are many risks and barriers women face to disclosing their experiences of violence and mental health problems which restrict women's access to psychological support. Culturally relevant services and trauma-informed interventions are necessary to respond to these issues. Service providers should be trained to effectively recognise and respond to survivors' mental health needs.

## INTRODUCTION

Domestic violence and abuse (DVA) is defined by the United Nations as 'a pattern of abusive behaviour toward an intimate partner

## STRENGTHS AND LIMITATIONS OF THIS STUDY

⇒ Kabul, Bamyan and Nangarhar were selected as study sites due to their diversity in ethnic groups, service provision and landscape (ie, urban and rural), and for the safety of the researchers travelling to these areas at the time of data collection.
⇒ All participants were purposively recruited to ensure an even split of genders.
⇒ The study only included female survivor participants who had already left their domestic situations prior to their involvement in the study and did not include women who were still experiencing domestic abuse for ethical reasons.
⇒ Data collection occurred in 2019—2 years before the Taliban seized control of Afghanistan. Our findings should, therefore, be interpreted within the context of Afghanistan's rapidly changing political landscape.

in a dating or family relationship, where the abuser exerts power and control over the victim'.[1] DVA is a highly prevalent form of violence against women (VAW) in Afghanistan, with half of all women estimated to have experienced some form of physical and/or sexual intimate partner violence (IPV) in their lifetime.[2 3] The prevalence of VAW in Afghanistan has likely been exacerbated by the decades of armed conflicts and humanitarian crises that the country has endured,[4] resulting in a range of compounding traumatic experiences for women. Survivors of VAW and armed conflict are more likely to experience a range of long-term mental health problems.[5 6] Indeed, Afghan citizens present a high prevalence of mental health problems including symptoms of Posttraumatic stress disorder (PTSD), depression and anxiety, with women exhibiting higher rates of psychological distress than men.[7]

The provision of certain mental health interventions has been effective at reducing the mental harms associated with VAW.[8] Psychosocial interventions are particularly effective in supporting the mental health of female survivors of IPV across a range of settings.[9] However, there is a severe paucity in the provision of mental health support in Afghanistan, which is largely supplied by non-government organisations (NGOs).[10 11] Where mental health services are available, many women still face barriers in accessing them due to the social stigmas associated with violence and mental health as well as the significant restrictions placed on women's freedoms. To improve the mental health support of female survivors, we must better understand how Afghan women experience violence both at an individual level to recognise their personal needs, and at a broader community level to gain a full picture of the barriers and challenges they face within their cultural and social environment.

## Study aims

The main aim of this study was to develop a set of recommendations in conjunction with the findings of a realist review on psychosocial interventions for survivors of IPV. Both the current study and the review were part of a broader National Institute for Health and Care Research (NIHR) Global Health Research Group[9] to develop a package of care for the mental health of women and children experiencing domestic violence in South Asia, which can be used by health service providers, NGOs and local communities to improve the mental health of female DVA survivors. The current study addresses the following research questions:

- ► How do women experience, conceptualise and cope with DVA and mental health?
- ► How do female survivors, male members of the community and service providers perceive and respond to DVA and mental health?
- ► What are the barriers and challenges faced by female DVA survivors around accessing mental health services and support?

## METHODS

## Patient and public involvement

The study team worked closely with a local partner NGO (who must remain anonymous for their safety but will be referred to in this paper as the pNGO) to ensure the needs of women affected by DVA were central to the study. The pNGO's staff were closely involved in the design of the study from its inception and led its implementation across all study sites with careful consideration of local dynamics, the safety of women involved in the interviews, and staff safety within the context of active nation-wide armed conflict. During data collection, the pNGO provided daily direct support to women who had experienced DVA and were strongly connected with their needs.

## Participants

The Afghan provinces of Kabul, Bamyan and Nangarhar were selected as study sites due to their diversity in ethnic groups, service provision and landscape (ie, urban and rural), and for the safety of the researchers travelling to these areas at the time of data collection.

Participants were recruited using purposive sampling across each of the three regions from three groups: (1) female DVA survivors; (2) male community members and (3) service providers who work with female DVA survivors. While female DVA survivors and service provider participants were recruited to understand women's narratives and experiences of violence, mental health and accessing support, male community member participants were recruited to provide further insight on community perspectives and responses to DVA, as well as information on the availability of mental health services.

Female survivor participants were identified through the existing networks of the pNGO in Afghanistan. Participants were considered eligible if they were no longer living with their abuser and assessed as not at risk of further violence as a result of their participation. In Kabul, participants were recruited through the pNGO's safe houses. In Bamyan and Nangarhar, participants were identified through the pNGO's network of women who had received legal or safe house services from closely affiliated organisations. Eligible survivors were approached by local current or former employees of the pNGO with whom they had existing relationships, to ask them if they wanted to participate in the study.

Male community member participants were recruited from several settings to ensure they represented a diverse range of socioeconomic characteristics, including university students, farmers, hospital clinicians and individuals living in an Internally Displaced People's camp. Most male participants were identified through the pNGO's informal networks. In the case of farmers, men were approached by the researcher.

Service provider participants were identified through the research team's informal networks and contacts and were further recruited via snowball sampling, where contacts approached other eligible candidates for participation in the study. Service provider participants were eligible if they worked with female DVA survivors. Researchers purposively selected participants to ensure an equal number of male and female service providers were included.

## Data collection

Topic guides were developed for service provider participants (see online supplemental document 1) and the female survivor and male community member participants (see online supplemental document 2). The topic guides were developed by the study team in close collaboration with the pNGO to ensure the local relevance of the questions asked. Questions were aligned with the broader goals of the project to develop a package of care to support the mental health of women and children

experiencing violence in Afghanistan. Additional questions about the conflict were asked to situate the resulting data in the current sociopolitical situation of the country. The topic guide was pilot tested previously as part of a smaller project with similar objectives conducted in 2017.[12]

Participant interviews were conducted in Dari/Pashto from April to October 2019. All participants gave their informed written or recorded verbal consent prior to interviews. Participants were told exactly how their stories would be used as part of the research and assured that the pNGO would not profit from sharing their story in any way. This was necessary as NGOs have been known to pay survivors to publish their stories as a means of attracting funding/donations without obtaining consent.

Semistructured interviews were conducted with female DVA survivors, either in a private room provided by the pNGO or in an alternative location chosen by the survivor. Survivor participants were asked about their personal experiences of violence, their mental health, details of coping mechanisms used to manage their mental health, mental health services they had used, responses to mental health in their community and the role of armed conflict in perpetuating violence and mental ill health.

Focus group discussions were conducted with male participants in settings that were easily accessible to them. Male participants were asked about their knowledge of mental health services, responses to mental health in their community, their views on the causes of and mental health consequences of VAW and the role of armed conflict in perpetuating violence and mental ill health.

Semistructured interviews were conducted with service provider participants in a private room at the service providers' clinic. Questions were asked about the types of violence and mental health problems they encountered in their work, how women and children were affected by violence, the role of armed conflict in perpetuating mental health problems, responses to mental health in the community and the role of mental health services.

All participant interviews were audiorecorded and pseudonymised with unique identification numbers before being transcribed verbatim and translated into English.

## Data analysis
A collaborative framework thematic analysis[13] was conducted in four stages: (1) preliminary themes were developed from a small sample of interview transcripts from each participant group (men, female survivors, service providers); (2) preliminary themes were discussed and used to develop a codebook with working theme definitions and example quotes; (3) transcripts were assigned randomly to four researchers and coded using the codebook, where each researcher reflexively adapted the codebook on a shared file and; (4) final themes from the codebook were merged and refined until all four researchers deemed the codebook to adequately reflect

the data as a whole. All transcripts were then reanalysed using the final agreed codebook.

## RESULTS
### Participant characteristics
60 female survivors, 60 male community members and 30 service providers were recruited for inclusion in the study. Table 1 shows the sociodemographic characteristics of participants.

### Armed conflict and VAW
All three participant groups were asked about their views on the impact of armed conflict on mental health. Many participants had experienced or witnessed traumatic incidents related to armed conflict, including the loss of loved ones, kidnappings, witnessing death, being caught in crossfire and seeing dead bodies. Participants also experienced forced internal displacement as a consequence of war, which in turn further exposed them to traumatic events, as outlined by one female survivor participant below.

> P20: We were forced to flee. Women had children in their arms as they fled through mountains and deserts […] people were running away. I saw a woman who gave birth on the way and throw her child in the river and run. I saw this with my own eyes. She was right in front of me. People were killed with bullets, houses were brought down with rockets. I saw a woman whose child was killed and she threw the dead body behind a wall and continued running. These are things I saw with my own eyes.

When exploring the association between armed conflict and VAW, the male community participants felt that the stress of witnessing and/or experiencing armed conflict affected men's mental health in ways that precipitated their use of physical VAW.

> P94: Violence increases inside and outside homes. People get into fights easily because of the mental toll of the war.

They also cited the widespread poverty following periods of armed conflict as another stress factor which contributed to the use of DVA by men: in Bamyan this was considered the main cause of DVA by participants.

> P97: There are no jobs and no money. Of course we develop mental health issues. […] We have experienced war and US bombings. Nobody can be at peace because they are expecting bombings.

Service providers similarly highlighted the ways in which war and poverty affect men's mental health and increased their use of violence towards women. They explained that mental health support for men was insufficiently accessible, leading to an inability to mitigate the traumatic effects of armed conflict and reduce subsequent violence.

**Table 1** Participant characteristics

| Participant group | Participant region | Participant ages | Participant professional/educational background | Participant gender |
|---|---|---|---|---|
| Female DVA survivors (n=60) | Kabul (n=20) Bamyan (n=20) Nangarhar (n=20) | 18–24 (n=26) 25–34 (n=26) 35–44 (n=6) >45 (n=2) | No formal schooling (n=25) Primary school (n=3.5) Middle school (n=6) High school (n=7) 12th grade/higher secondary (n=10) Tertiary (n=9) Vocational (n=0) | Female (n=60) |
| Male members of the community (n=60) | Kabul (n=30) Bamyan (n=30) Nangarhar (n=30) | | Internally displaced men (n=10) Labourers and farmers (n=10) University students and activists (n=20) Hospital workers (N=20) Teachers (n=10) Government officials (n=10) | Male (n=60) |
| Service providers (n=30) | Kabul (n=10) Bamyan (n=10) Nangarhar (n=10) | 18–24 (n=15) 25–34 (n=12) 35–44 (n=1) >45 (n=2) | NGO psychologist (n=6) NGO counsellor (n=6) Academic or NGO researcher (n=5) Hospital psychologist (n=4) NGO worker (n=4) NGO medical worker (n=2) Addiction clinic worker (n=1) Awareness raising (n=1) Doctor (n=1) | Male (n=16) Female (n=14) |

DVA, domestic violence and abuse; NGO, non-government organisation.

P28: We have to consider men that have mental health issues. They create problems and they have to be treated. The community should help identify such people. Men are sadistic and paranoid and hurt their daughters and wives. This awareness should be spread.

Reasons given for this lack of support included a dearth of formal support services, a reliance on traditional healers and religious leaders for spiritual support, and the widespread stigma around mental health in the general public.

The female survivor participants explained how experiencing armed conflict led to them developing symptoms of poor mental health such as nightmares, emotional detachment and persistent fear. However, the female survivor participants almost always perceived the violence perpetrated against them by their husbands or families as being more significant to their psychological health than their experiences of armed conflict.

### Conceptions and narratives around VAW

Participants from all three groups spoke of the high prevalence of VAW in Afghanistan, which they all reported witnessing in their homes or local communities.

The male participants openly shared that they knew men who beat their wives and that this was a common occurrence across households. However, the acceptability of VAW was contested, where some men expressed views that VAW was unacceptable under any circumstance while others felt that the occasional use of VAW was justifiable.

Violence was usually seen as justifiable in instances where women attempted to transcend social expectations (eg, by leaving the house without permission) with violence being used against them as a means of teaching them about their place in society. While the male participants understood what was meant by the term violence, many did not identify their or other men's use of VAW as being 'violent', as the action was perceived as warranted.

P98: If she behaves badly or does something bad, of course you have to beat her. If she doesn't do anything wrong, why would anyone hit her?

The female survivor participants disclosed encountering a range of typologies of DVA, most commonly physical abuse. Participants explored how gender norms in Afghan society facilitated this abuse through the sociocultural normalisation of VAW which occurred in both public and private spaces, where any bystanders witnessing the abuse would rarely intervene. This acceptability of VAW was compounded by women's limited social agency which resulted in the men in their lives (ie, husbands and male relatives) being responsible for making important decisions for them, and leaving them to feel that their own lives had little value in society.

While the majority of participants' accounts of DVA involved physical violence from husbands or other family members, both the male and female survivor participant groups referred to the role of the mother-in-law as perpetrators of DVA and enforcers of restrictive gender norms for women. This behaviour was

described as being enacted both through perpetrating their own abuse and through instigating their sons to use violence against their wives. Female survivor participants recounted feeling psychologically impacted by experiencing violence from another woman, particularly given the woman is likely to have been a victim of the same violence herself.

P49: I mainly blame my mother-in-law and sister-in-law, they were the ones who provoked my husband and made him suspicious and even forced him to beat me.

Additionally, all three participant groups perceived violence as a common component in child marriages, as when a girl is forced into marriage, she is subsequently considered a woman despite her young age, and thus subject to the same sociocultural expectations as adult women. When she does not adhere to these expectations, she is blamed and submitted to abuse.

These entrenched gender norms around the acceptability of violence were more contested among service providers, where some condoned the use of VAW under certain circumstances…

P57: When you ask the girl her issues and stories, at times it is her fault because she doesn't listen to her husband. For example, he forbids her from going to her parents' home, but she goes anyway and it's her mistake. We tell the girl that she shouldn't have spoken out, shouldn't have reacted and should have been patient.

While others were completely against it.

P21: Unfortunately, we see that doctors and other professionals blame the victim. This is common.

Experiences of sexual violence, forced prostitution, forced labour (eg, being forced to quit school to perform household duties) and threats of murder (eg, honour killings) were also described by female survivor participants. Different typologies of violence were rarely experienced in isolation, with women recounting experiencing multiple types of violence throughout their lifetime.

### Conception and narratives around mental health

Female survivor participants described experiencing a range of mental health impacts as a result of the violence they had endured, including effects on their sense of identity, their belongingness to their family and society, and their overall well-being. In many cases, survivors and service providers described psychological impacts which aligned with the biomedical characteristics of diagnosed mental health disorders, using words such as anxiety and depression to describe how they felt.

P28: The major impact of every kind of violence is its mental impact. Women have [Generalised Anxiety Disorders] from the constant abuse they experience…

However, participants also used other phrases that did not align with diagnostic categories such as 'thinking a lot' or 'isolating oneself'.

All three participant groups directly referenced or indirectly alluded to the culture of silence, stigma and taboo surrounding mental health in Afghan society, as well as the cultural norms relating to a reluctance or hindrance in sharing suffering. Participants described experiences of shame, harassment, ostracisation or subjection to further violence as common consequences of sharing mental health problems with others.

P28: They see psychological issues as madness. This shows our cultural problems. Everyone is afraid of being labelled.

The male participants identified structural factors such as armed conflict and poverty as key contributors to mental ill health for men. When discussing factors which affect women's mental health, the male participants often attributed women's 'overly emotional natures' as the main factor leading to their own mental suffering. Conversely, the service provider participants cited social inequality, gender norms and violence as the leading factors contributing to women's mental ill health.

When discussing the ways in which violence and mental health may be linked, the male and female survivor participants believed it was mainly the physical symptoms of violence (eg, brain damage from being physically attacked) that caused mental health problems in survivors. This merging of physical and psychological understandings of mental health further coincided with how participants described women with mental health issues as physically expressing their distress (eg, by pulling their hair, experiencing headaches).

When exploring conceptualisations of good mental health, female survivor participants often described a lifestyle, such as living the life you choose, instead of a particular mental state. Service provider participants similarly emphasised the importance of seeing the meaning in life, as well as taking care of one's physical health.

### Coping strategies and support

Both female survivor and service provider participants described the importance of utilising coping strategies in order to manage their mental health. These are summarised in figure 1.

### Service provision

All participants identified a lack of service provision to address gender-based violence and mental ill health. In Bamyan, service provider participants said there were no mental health services in the region. In other regions, services such as cognitive–behavioural therapy were mentioned but were unable to meet the high demand of people needing to access them. Both the male and female survivor participant groups similarly reported a lack of adequate mental health services.

**Coping strategies utilised by female survivors of violence and abuse**

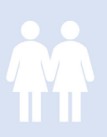

**Keeping busy with everyday tasks**
- Cooking
- Reading books
- Listening to music
- Exercising

*"I work in the kitchen all day to keep my mind off"*

*"I busy myself with books and I also sew clothes"*

*"I read books or listen to music"*

*"I feel happy when we exercise", "I take a walk to take the burden off my mind"*

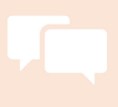

**Talking to others**
- Talking to family
- Peer support from friends
- Talking to counsellors (often in Safe Houses)

*"If those families and community work with the mentally ill they will recover quicker"*

*"I have a close friend here that I always talk to and feel good after sharing my pain"*

*"If you want to lighten your heart, you should talk to the psychologist"*

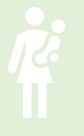

**Spending time with their children**
- Focusing on their children as a source of pleasure and accomplishment
- Finding content in their children's happiness

*"I play with my children and forget everything"*

*"I busy myself with my children. I study with them or put them to sleep. Or we walk around outside"*

*"I feel good that my children are studying, that they are happy and healthy"*

*"My daughter makes me happy. I am grateful that I at least have her with me"*

**Figure 1** Coping strategies used by female survivors of violence and abuse.

The most common type of support described by service provider and female survivor participants was counselling, where delivery focused on helping women come to terms with their role in society as a means of reconciling their mental health with the violence they experienced. In cases where counselling was not considered effective, service users were often sent to medical doctors for treatment. References to the provision of psychosocial interventions such as cognitive behavioural therapy were rarely described.

Many of the female survivor participants had access to psychologists and counsellors through the services provided by the safe house where they were staying in or the civil society organisations they were using. Survivors' views on safe houses and the psychological support offered were variable. Some described feeling distressed at being surrounded by other women with similar experiences and having to relive their own experiences of DVA.

Interviewer: How do you feel when you listen to stories of other women [at the safe house] who have suffered?

P17: Honestly, I don't listen to other women's stories. When I feel like it will make me sad I just leave immediately. I never listen to their stories. I still don't know the cases of other women. I don't ask them any questions.

P2: I don't talk to the psychologist here so much because I remember my past and my condition worsens.

Meanwhile others described them as positive spaces where they could enjoy a collegial environment and access support from psychologists with whom they could share their suffering.

Interviewer: Is there anyone you can talk to about [your experiences]?

P75: I talk to the staff. I feel better then […] I like to talk to everyone.

P5: [the psychologist] helps me a lot. I even take my legal issues to her. She has worked here for a long time so she understands our cases. She comforts and assures me that my case will be resolved.

Interviewer: Would you recommend the psychologist to others?

P17: Yes, I do. When I see girls crying constantly, I tell them to go. Some of them agree.

Interviewer: Why would you recommend them?

P17: Because she says good things. She keeps your secrets. Other girls don't. If you want to lighten your heart you should talk to the psychologist.

## DISCUSSION

Our findings indicate that many women in Afghanistan have experienced multiple and compounding traumatic experiences of violence and complex, competing psychosocial concerns. These experiences are precipitated by social stigma and gender norms which, combined with a

lack of effective mental health services, restrict women's abilities to access support. Services themselves are also currently incredibly constrained. Under Taliban rule, NGOs can no longer provide violence-specific support, and women are forbidden from working for NGOs which in turn curtails their access to services.

Despite these limitations, settings which can be used to feasibly improve the well-being of women in Afghanistan have been identified.[14] Additionally, Mental Health and Psychosocial Support (MHPSS) is one of the few health agendas that are still allowed to operate in Afghanistan, presenting spaces to deliver support. We have contextualised our findings within the current constraints on services—and findings from the broader NIHR Global Health Research project[9]—to develop a set of recommendations, of which elements can be incorporated into existing structures to shape interactions and improve the support of female survivors (see table 2).

These recommendations are also relevant beyond Afghan settings and may be adopted by organisations, services and communities in other countries affected by humanitarian crises, armed conflict and high rates of VAW. Indeed, many of these recommendations have already been piloted in one or more of these settings. For instance, Telehealth and e-Health approaches to mental health have been shown to increase service access to remote areas in Afghanistan[15] and reduce stigma. Although studies have focused on mental health more broadly and not violence survivors specifically, given the instability and remoteness of rural populations in Kabul where the majority of healthcare services are located, e-Health could be a simple and low-cost approach. However, effective e-Health mental health interventions rely on specialist knowledge which may pose problems given there is still a lack of trained mental health professional availability in Afghanistan.

Capacity building was highlighted as a priority area by service provider participants who highlighted this need for more trained specialists who hold a deep understanding and empathy for survivors' needs. In addition, they described the mechanisms by which the lack of public awareness around mental health and violence not only reduces rates of disclosure amongst the general public but also acts as a barrier to community and familial support for survivors due to the stigma associated with mental health problems and domestic abuse. In the absence of interventions targeted at reducing stigma, the perspectives explored by our service provider participants align with existing evidence suggesting services should create private and confidential spaces to quell survivors' fears of identification, increase survivor confidence in both the intervention and service provider[16–18] and improve their overall uptake of services.[19–21] These recommendations may be feasibly implemented by providers across a range of settings including physicians, nurses and volunteers.

Furthermore, the role of the family, both as perpetrators of abuse (eg, partner, mother-in-law or other family members) and as actors of support (eg, forging positive relationships with family members or spending time with children as a source of joy for survivors), suggests that family-focused interventions may improve survivors' access to support. Although Afghan women who experience IPV may access family planning services less frequently,[2] along with maternity care services they may act as potential sites to support pregnant women or mothers experiencing IPV.[22] The incorporation of social support, mother and child, and mind-body interventions in healthcare service delivery may also be a simple and low-cost solution, given many participants used elements of these within their personal coping mechanisms.

## Strengths and limitations

This study explores personal narratives around violence and mental health across three ethnically, linguistically and geographically diverse regions in Afghanistan, addressing the systemic need for improvement in the provision of support to survivors.

One limitation of this study is that all female survivor participants had left their domestic situations before their involvement in the study. While this was done to ensure participants' safety, our findings may not entirely account for the needs of DVA survivors who are still living in their abusive household.

Additionally, in 2021, the Taliban seized control of the whole of Afghanistan, resulting in the further systematic erosion of women's rights across the country. This took place after the completion of the data collection stage, which will have subsequently affected the experiences and needs of female survivors, as well as the state of mental health service provision across the country. It is, therefore, important that the recommendations we present in this study are adapted within the context of Afghanistan's rapidly changing political landscape.

## Implications for clinicians and policy-makers

Clinicians, service providers and volunteers in Afghanistan should be provided with training around effectively working with survivors of violence and mental health problems in a culturally appropriate manner. Service providers who are able to still operate in Afghanistan, such as MHPSS providers, may wish to use the training resource developed from the findings of this study to increase service providers' knowledge and confidence when supporting survivor service users in their care.[23]

Training should emphasise the importance of privacy and confidentiality and consciously tackle the harmful biases that service providers may have which stigmatise female survivors. Future studies may wish to pilot and evaluate interventions which aim to reduce the stigma of mental health in service providers in Afghanistan.

Additionally, trauma-informed and survivor-centred mental health interventions should be incorporated into service provision where possible. Clinicians and service providers should also be attentive to the fact that women may have experienced violence from more than one perpetrator within the household and promote the

**Table 2** Recommendations for services and interventions for female survivors of violence, abuse and mental health problems

| Recommendation | Description |
| --- | --- |
| Adopting trauma informed IPV tailored approaches | ▶ Tailoring services and interventions to meet the individual circumstances and needs of survivors.<br>▶ Adopting trauma-informed approaches and IPV tailored psychological treatments in care settings to meet women's complex needs.[18 24–26]<br>▶ Including strategies which enable survivors to manage their trauma symptoms within interventions.[18 24 27 28] |
| Prioritising privacy and confidentiality (in service provision settings and through service provision engagements) | ▶ Ensuring interactions between service providers and survivors prioritise privacy and confidentiality by conducting interventions in private and secure settings.[19–21]<br>▶ Ensuring service providers actively assure confidentiality to survivors to support disclosures by:<br> – Explaining to survivors why disclosures are beneficial.<br> – Respectfully and non-judgementally discussing IPV without asking potentially stigmatising questions.<br> – Making use of open-ended and behaviourally anchored questions using a broad understanding of abuse (including physical, sexual and emotional forms).<br> – Ensuring multiple opportunities for disclosure are provided over time.<br> – Using active listening and empathetic approaches to build trust and rapport.[19 25 26 29–31] |
| Developing targeted training for service providers | ▶ Implementing context-specific and specialised training programmes for service providers and clinical staff around responding to domestic violence and mental health problems. In settings such as Afghanistan where interventions for violence are forbidden, the focus can be placed on improving the emotional challenges the woman is facing as a gateway to opening up the discussion around violence.<br>▶ Addressing stigmas and negative stereotypes around domestic violence and mental health in service provider training resources.[32] |
| Including social support, mother and child, and mind-body interventions | ▶ Implementing interventions in which women's informal social support structures are strengthened to enhance their ability to cope, access resources and improve their psychological well-being.[18 33]<br>▶ Supporting mother and child interventions to improve the mental health of female survivors of violence through strengthening the bond between mother and child. This provides women with a distraction from the stressors of their situation through engaging in play, and providing women with motivation, purpose, joy and love.[25 34 35]<br>▶ Implementing interventions which engage the body in healing from trauma (eg, yoga, meditation, somatic experiencing) which can support survivors in their ability to manage symptoms and triggers,[36] increase their quality of life and improve psychological outcomes.[18 37 38] |
| Including family-focused interventions | ▶ Being attentive within interventions and services to the varied identities of perpetrators of violence, who are not only comprised of intimate partners but can also be other members of the family, including mother in-laws.<br>▶ Incorporating approaches which address mother-in-law violence through understanding the mother-in-law as a 'victim' of gender-based violence and aiming to resolve the trauma she has experienced herself as a first step to addressing her use of violence against others.<br>▶ Implementing interventions within services which strengthen positive trauma-informed relationships between women and mother-in-laws to break the cycle of violence within households. This will prevent future violence, strengthen social support systems for women and allow for the possibility of reconciliation and/or therapeutic processes to take place within households where violence has occurred.[39] |
| Increasing awareness raising and engagement activities on violence against women and mental health | ▶ Raising awareness about the psychosocial impact of VAW and associated mental health challenges in Afghanistan to reduce stigma, improve access to services, and cultivate supportive community environments.<br>▶ Community mobilisation and awareness through culturally relevant adaptations of advocacy interventions, such as social media and outreach campaigns which target harmful stereotypes and the stigma associated with gender-based violence and mental health[20 28 40 41] driven by grass roots, NGO and governmental sources. |
| Considering alternative delivery formats such as mHealth or Telehealth | ▶ Implementing trauma-informed Telehealth, e-Health or mHealth approaches to mental health services to reduce stigma and increase service access to remote, urban or displaced communities in Afghanistan.<br>▶ Ensuring the delivery of alternative formats Telehealth, e-Health or mHealth account for the risks of disclosure to survivors such as further violence or stigmatisation in the community.<br>▶ Connecting primary care workers in rural and remote areas with specialists to provide better access to care.[15] |

IPV, intimate partner violence; NGO, non-government organisation; VAW, violence against women.

use of family-focused interventions which address all members of the household and not just couples. Women would further benefit from implementing social support, mother and child, and mind-body interventions within services. Future research should explore the efficacy and feasibility of implementing telehealth mental health interventions specifically for female survivors of violence who face risks and barriers in disclosing violence and their mental health problems.

Participants expressed a desire for increased advocacy and awareness efforts at a local and government level to tackle the stigma associated with violence and mental health. Policy-makers and NGOs should, therefore, explore culturally relevant methods of addressing these harmful social norms.

## Conclusion

VAW and the associated negative mental health outcomes are highly prevalent in Afghanistan, and women face many barriers to disclosing these experiences, including social stigma and restrictions to their freedom. Culturally relevant trauma-informed interventions are necessary to respond to these issues. Service providers should be trained to effectively recognise and respond to survivors' needs.

**Contributors** Study conception and design: JM, AA, LS and DD. Analysis and interpretation of results: JM, AA, LS, SAP and AK. Writing–original draft: AK. Writing–review and editing: JM, AK, SAP, AA and DD.JM is responsible for the overall content as guarantor.

**Funding** This study was funded by the NIHR Global Health Research Group on Violence and Mental Health in South Asia (VAMHSA) (grant reference: 17/63/47) using UK aid from the UK Government to support global health research and The UKRI Violence, Abuse and Mental Health Network (VAMHN) (grant ref: ES/S004424/1).

**Disclaimer** The views expressed in this publication are those of the authors and not necessarily those of the NIHR, the UK Department of Health and Social Care or UKRI.

**Competing interests** None declared.

**Patient and public involvement** Patients and/or the public were involved in the design, or conduct, or reporting, or dissemination plans of this research. Refer to the Methods section for further details.

**Patient consent for publication** Not applicable.

**Ethics approval** This study involves human participants and ethical approval was granted by the UCL Research Ethics Committee (ethics approval reference: 2744/007); the London School of Hygiene & Tropical Medicine Research Ethics Committee (Ethics approval reference: 22818) and the Islamic Republic of Afghanistan, Ministry of Public Health (Ethics approval reference: IRB.1902.0007). Participants gave informed consent to participate in the study before taking part.

**Provenance and peer review** Not commissioned; externally peer reviewed.

**Data availability statement** Data sharing not applicable as no datasets generated and/or analysed for this study.

**ORCID iDs**
Anjuli Kaul http://orcid.org/0000-0002-5637-5536
Jenevieve Mannell http://orcid.org/0000-0002-7456-3194

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
