## [Reviewer comments · BMJ Open]

ARTICLE DETAILS

TITLE (PROVISIONAL)	What are the experiences and psychosocial needs of female survivors of domestic violence in Afghanistan? A qualitative interview study in three Afghan provinces
AUTHORS	Kaul, Anjuli; Saboor, Lamba; Ahmad, Ayesha; Mannell, Jenevieve; Paphitis, Sharli; Devakumar, Delan

VERSION 1 – REVIEW

REVIEWER	Baird, Kathleen University of Technology Sydney
REVIEW RETURNED	05-Mar-2024

GENERAL COMMENTS	Paper is well written, has a logical flow. It explores an important topic and in provides an insight into a country which is rarely researched in light of the topic being explored. The methodology chosen is suitable for the study aims. Ethical principles are considered. However, I do have some questions about the study. Introduction Page 3. It would be helpful to include a definition of DVA. Methods Page 4 - Participants it is not clear why male participants were recruited, what additional information was gleaned about their level of knowledge if mental health and referral services and why 60? Requires an explanation – why so many males are recruited for a study on female experiences? Also was it not a conflict of interest if the male farmers who participated in the focus groups were selected by the researcher? Page 5 - Data analysis – armed conflict and VAW – only one quote was used, and it is not identified whether the quote is a female or male. To support the theme, more than quote should be used. I note that statements have been made about women thoughts and views, but there is a lack of quotes confirming the propositions. For example, page 9, 2nd paragraph. Discussion Page 9 – 4th paragraph line 34, the last statement of the study being conducted as a part of a the broader NIHR global health research should be included as an opening statement in the introduction of the paper. Line 52 – remove the use of the word etc, not appropriate in a
--

	journal – use the term other family members instead. There or very limited discussion on the data from the NGO’s and how this is used to formulate the discussion or recommendations for service and interventions for female survivors if violence, abuse, and mental health problems.
--	---

VERSION 1 – AUTHOR RESPONSE

Reviewer 1: Dr. Kathleen Baird, University of Technology Sydney

- 4) **Reviewer’s comments to the Author:** Paper is well written, has a logical flow. It explores an important topic and in provides an insight into a country which is rarely researched in light of the topic being explored. The methodology chosen is suitable for the study aims. Ethical principles are considered. However, I do have some questions about the study.
Author’s response: On behalf of the authorship team we would like to sincerely thank Dr Baird for their time spent reading our manuscript and for their helpful, constructive and insightful comments. We are very grateful for their expertise and for their comments highlighting the importance of this subject.

Introduction

- 5) **Reviewer’s comments:** Page 3. It would be helpful to include a definition of DVA.
Author’s response: Thank you for this comment. We have added a definition of DVA to the first sentence of the introduction. This section now reads: *Domestic violence and abuse (DVA) is defined by the United Nations as “a pattern of abusive behaviour toward an intimate partner in a dating or family relationship, where the abuser exerts power and control over the victim. (1)”*

Methods

- 6) **Reviewer’s comments:** Page 4 - Participants it is not clear why male participants were recruited, what additional information was gleamed about their level of knowledge if mental health and referral services and why 60? Requires an explanation – why so many males are recruited for a study on female experiences? Also was it not a conflict of interest if the male farmers who participated in the focus groups were selected by the researcher?
Author’s response:
Male participants were recruited to provide further insight into community perspectives and responses to DVA, as well as information on the availability of mental health services. Due to the large role that stigma and cultural norms play in the perpetuation of DVA and VAW, we considered it important to hear from male participants also, particularly given the centrality of men as decision-makers in the Afghan context. In terms of the sample size, we intended to interview 20 participants from each province (n=60 across the 3 study provinces) to help draw conclusions when comparing results from different sites. To clarify the reason we chose to interview male participants within the manuscript, we have added a sentence under the second paragraph of the “Participants” section which now reads: *“Participants were recruited using purposive sampling across each of the three regions from three groups: i) female DVA survivors; ii) male community members; and iii) service providers who work with female DVA survivors. Whilst female DVA survivors and service provider participants were recruited to understand women’s narratives and experiences of violence, mental health and accessing support, male community member participants were recruited to provide further insight on community perspectives and responses to DVA, as well as information on the availability of mental health services.”*
- 7) **Reviewer’s comments:** Page 5 - Data analysis – armed conflict and VAW – only one quote was used, and it is not identified whether the quote is a female or male. To support the theme, more than quote should be used.
Author’s response: We have added the gender of the participant to the text preceding the quote under the “Armed conflict and VAW section”. This now reads:
Participants also experienced forced internal displacement as a consequence of war, which in turn further exposed them to traumatic events, as outlined by one female survivor participant below.

P20: "We were forced to flee. Women had children in their arms as they fled through mountains and deserts [...] people were running away. I saw a woman who gave birth on the way and throw her child in the river and run. I saw this with my own eyes. She was right in front of me. People were killed with bullets, houses were brought down with rockets. I saw a woman whose child was killed and she threw the dead body behind a wall and continued running. These are things I saw with my own eyes."

Additionally, we have added three additional quotes throughout the armed conflict section from the male participants and service provider participants to support the text (see quotes added from P94, P97 and P28). These read:

P94: "Violence increases inside and outside homes. People get into fights easily because of the mental toll of the war."

P97: "There are no jobs and no money. Of course we develop mental health issues. [...] We have experienced war and US bombings. Nobody can be at peace because they are expecting bombings."

P28: We have to consider men that have mental health issues. They create problems and they have to be treated. The community should help identify such people. Men are sadistic and paranoid and hurt their daughters and wives. This awareness should be spread.

- 8) **Reviewer's comments:** I note that statements have been made about women thoughts and views, but there is a lack of quotes confirming the propositions. For example, page 9, 2nd paragraph.

Author's response: Thank you for flagging this. We have added quotes to support the proposition highlighted. This section now reads:

Many of the female survivor participants had access to psychologists and counsellors through the services provided by the safe house they where they were staying in or the civil society organisations they were using. Survivors' views on safe houses and the psychological support offered were variable. Some described feeling distressed at being surrounded by other women with similar experiences and having to relive their own experiences of DVA.

Interviewer: "How do you feel when you listen to stories of other women [at the safe house] who have suffered?"

P17: "Honestly, I don't listen to other women's stories. When I feel like it will make me sad I just leave immediately. I never listen to their stories. I still don't know the cases of other women. I don't ask them any questions."

P2: "I don't talk to the psychologist here so much because I remember my past and my condition worsens."

Meanwhile others described them as positive spaces where they could enjoy a collegial environment and access support from psychologists with whom they could share their suffering.

Interviewer: "Is there anyone you can talk to about [your experiences]?"

P75: "I talk to the staff. I feel better then [...] I like to talk to everyone."

P5: [the psychologist] helps me a lot. I even take my legal issues to her. She has worked here for a long time so she understands our cases. She comforts and assures me that my case will be resolved."

Interviewer: "Would you recommend the psychologist to others?"

P17: "Yes, I do. When I see girls crying constantly, I tell them to go. Some of them agree."

Interviewer: "Why would you recommend them?"

P17: "Because she says good things. She keeps your secrets. Other girls don't. If you want to lighten your heart you should talk to the psychologist."

Discussion

- 9) **Reviewer's comments:** Page 9 – 4th paragraph line 34, the last statement of the study being conducted as a part of a the broader NIHR global health research should be included as an opening statement in the introduction of the paper.

Author's response: We have moved this statement to the introduction under the study aims section. This section now reads:

The main aim of this study was to develop a set of recommendations in conjunction with the findings of a realist review on psychosocial interventions for survivors of IPV. Both the current study and the review were part of a broader NIHR Global Health Research Group (9) to develop a package of care for the mental health of women and children experiencing domestic violence in South Asia, which can be utilised by health service providers, NGOs and local communities to improve the mental health of female DVA survivors. The current study addresses the following research questions:

- How do women experience, conceptualise and cope with DVA and mental health?
- How do female survivors, men and service providers perceive and respond to DVA and mental health?
- What are the barriers and challenges faced by female DVA survivors around accessing mental health services and support?

The section on page 9 referenced in the reviewer’s original comment now reads:

“Additionally, Mental Health and Psychosocial Support (MHPSS) are one of the few health agendas that are still allowed to operate in Afghanistan, presenting spaces to deliver support. We have contextualised our findings within the current constraints on services – and findings from the broader NIHR global health research project (8) - to develop a set of recommendations in conjunction with the findings from the broader, of which elements can be incorporated into existing structures to shape interactions and improve the support of female survivors (see Table 2).”

- 10) **Reviewer’s comments:** Line 52 – remove the use of the word etc, not appropriate in a journal – use the term other family members instead.

Author’s response: We have removed the term “etc.” and replaced with the term “other family members” as advised. This sentence now reads: *Additionally, the role of the family, both as perpetrators of abuse (e.g. partner, mother-in-law or other family members) and as actors of support...*

- 11) **Reviewer’s comments:** There or very limited discussion on the data from the NGO’s and how this is used to formulate the discussion or recommendations for service and interventions for female survivors if violence, abuse, and mental health problems.

Author’s response: We have reworked the discussion to include a more targeted exploration of how the views of our service provider participants have been used to formulate the recommendations. This section in the discussion now reads:

Capacity building was highlighted as a priority area by service provider participants who highlighted the need for more trained specialists who hold a deep understanding and empathy for survivors’ needs. In addition, they described the mechanisms by which the lack of public awareness around mental health and violence not only reduce rates of disclosure amongst the general public, but also act as a barrier to community and familial support for survivors due to the stigma associated with mental health problems and domestic abuse. In the absence of interventions targeted at reducing stigma, the perspectives explored by our service provider participants align with existing evidence suggesting services should create private and confidential spaces to quell survivors’ fears of identification, increase survivor confidence in the intervention and service provider, (15-17) and improve their overall uptake of services. (18-20) These recommendations may be feasibly implemented by providers across a range of settings including physicians, nurses, and volunteers.

VERSION 2 – REVIEW

REVIEWER	Baird, Kathleen University of Technology Sydney
REVIEW RETURNED	04-May-2024
GENERAL COMMENTS	Happy with changes made, the authors have responded well to the issues raised in the previous review.